# Historic and recent trends in county-level coronary heart disease death rates by race, gender, and age group, United States, 1979-2017

**Adam S. Vaughan**⬤*, **Linda Schieb, Michele Casper**

Division for Heart Disease and Stroke Prevention, Centers for Disease Control and Prevention, Atlanta, GA, United States of America

* avaughan@cdc.gov

**Data Availability Statement:** All relevant data are within the manuscript and its Supporting Information files.

## Abstract

Given recent slowing of declines in national all-cause, heart disease, and stroke mortality, examining spatiotemporal distributions of coronary heart disease (CHD) death rates and trends can provide data critical to improving the cardiovascular health of populations. This paper documents county-level CHD death rates and trends by age group, race, and gender from 1979 through 2017. Using data from the National Vital Statistics System and a Bayesian multivariate space-time conditional autoregressive model, we estimated county-level age-standardized annual CHD death rates for 1979 through 2017 by age group (35–64 years, 65 years and older), race (white, black, other), and gender (men, women). We then estimated county-level total percent change in CHD death rates during four intervals (1979–1990, 1990–2000, 2000–2010, 2010–2017) using log-linear regression models. For all intervals, national CHD death rates declined for all groups. Prior to 2010, although most counties across age, race, and gender experienced declines, pockets of increasing CHD death rates were observed in the Mississippi Delta, Oklahoma, East Texas, and New Mexico across age groups and gender, and were more prominent among non-white populations than whites. Since 2010, across age, race, and gender, county-level declines in CHD death rates have slowed, with a marked increase in the percent of counties with increasing CHD death rates (e.g. 4.4% and 19.9% for ages 35 and older during 1979–1990 and 2010–2017, respectively). Recent increases were especially prevalent and geographically widespread among ages 35–64 years, with 40.5% of counties (95% CI: 38.4, 43.1) experiencing increases. Spatiotemporal differences in these long term, county-level results can inform responses by the public health community, medical providers, researchers, and communities to address troubling recent trends.

**Funding:** The authors received no specific funding for this work. The findings and conclusions in this report are those of the authors and do not necessarily represent the official position of the Centers for Disease Control and Prevention.

**Competing interests:** The authors have declared that no competing interests exist.

## Introduction

In 2017, coronary heart disease (CHD) was the underlying cause of over 360,000 deaths in the United States (US), representing about 60% of heart disease deaths [1, 2]. Despite the stagnation in national heart disease death rates since 2010 [3, 4], national coronary heart disease (CHD) death rates have continued to decline [1]. This recent decline occurs following a long period of differences in CHD mortality trends by age, race, gender, and geography. National trends in CHD mortality by race and gender diverged in the 1970s [5, 6], with slower declines for blacks than whites and for women than men. In the 1990s, declines in CHD mortality slowed among younger adults, but accelerated in older adults [7, 8]. Since 2000, CHD death rates continued decline, with evidence of stagnation of declines in younger women [9].

However, national trends in CHD mortality may obscure critical differences among US counties [10–12]. Although heart disease and CHD death rates have historically declined more slowly in the southern United States [6, 11–13], recent increases in heart disease death rates have been spread throughout the country, especially among adults ages 35–64 [10, 11]. Given this geographic variation and recent slowing of national heart disease death rates [1, 3, 4, 10], fully documenting the spatiotemporal distributions of CHD death rates and trends can provide public health professionals, clinicians, researchers, and communities with data critical to improving the cardiovascular health of populations. Therefore, this paper documents county-level CHD death rates and trends by age group, race, and gender from 1979 through 2017.

## Methods

### CHD mortality data

For US residents ages 35 and older, we obtained county-level annual counts of CHD deaths from the National Vital Statistics System of the National Center for Health Statistics (NCHS) from 1979–2017. Over this study period, the classification of race and ethnicity changed nationally and within states, restricting our study population to black, white, or other [14]. Hispanic ethnicity was not recorded on death certificates from all states for the duration of the study period. We included deaths for which the underlying cause of death was CHD according to the 9th, and 10th revisions of the International Classification of Diseases (ICD) (ICD–9: 410–414, 429.2; ICD-10: I20–I25). This definition of ICD codes permitted a consistent comparison of CHD deaths across this 39 year period [15].

The unit of analysis was the county. Given changes in county definitions during the study period (e.g., the creation of new counties), a single set of 3,115 counties based on the most recent county definitions was used for the entire study period. We used NCHS bridged-race estimates for annual county-level populations [16].

### Estimating death rates

We estimated US county-level CHD death rates for the years 1979 through 2017 using a Bayesian multivariate space-time conditional autoregressive model. With this model, we estimated rates for each year and county by age group (ages 35 and older, ages 35–64, and ages 65 and older), race (black, white, other), and gender (men, women). Rates for race and gender included only individuals aged 35 years and older. Adults aged 35 and older comprise 99.7% of CHD deaths. Additionally, the age groups included in this analysis have recently experienced similar county-level trends in heart disease and stroke mortality and have been identified as key populations by national initiatives that seek to prevent cardiovascular disease events [10–12, 17, 18].

Details of this model have been previously described [19] and used to model heart disease, heart failure, and stroke death rates [11, 12, 17]. This model is based on the popular Besag-York-Mollié conditional autoregressive model for spatially-referenced count data [20] and incorporates correlation across space, time, and demographic groups. By iteratively estimating parameters and borrowing strength from neighbors, these models generate more precise, reliable rates even in the presence of small case counts and small populations [19, 21]. We fit this model with a Markov chain Monte Carlo (MCMC) algorithm using user-developed code in the R programming language. All death rates were age-standardized to the 2000 standard US population using 10-year age groups.

Specifically, we modeled $Y_{ikt}$, the number of deaths due to heart disease in county i and age group k during year t from a population of size $n_{ikt}$, using a Poisson distribution of the form $Y_{ikt} \sim Pois(n_{ikt}\lambda_{ikt})$, where $\lambda_{ikt}$ denotes the heart disease mortality rate. To model $\lambda_{ikt}$, we assume $ln(\lambda_{ikt}) \sim N(\beta_{kt} + Z_{ikt}, \tau_k^2)$, where $\beta_{kt}$ is a random intercept for each group for each year with a vague N(0,100) prior, $Z_{ikt}$ is a spatiotemporal random effect that incorporates correlation between groups, and $\tau_k^2$ is a variance parameter with a weakly informative gamma prior [22].

To account for these various sources of dependence, we model $Z_{ikt}$ using a multivariate space-time conditional autoregressive (MSTCAR) model [19], a special case of the multivariate CAR model of Gelfand and Vounatsou [23]. This MSTCAR model shrinks the random effects for each county toward the values in neighboring counties. Similarly, temporal structure is accounted for within the MSTCAR model by shrinking estimates toward adjacent years using a standard autoregressive order 1 (AR(1)) model with a beta prior. Finally, correlations between groups are estimated via an unstructured covariance matrix with an inverse Wishart prior [22].

We ran the MCMC algorithm with four chains for 6000 iterations, diagnosing convergence via trace plots for many of the model parameters and discarding the first 3000 iterations as burn-in. We generated estimates based on posterior medians, and 95% credible intervals were obtained by taking the 2.5- and 97.5-percentiles from the thinned post-burn-in samples.

## Estimating trends in death rates

For each demographic group within each county, we measured temporal trends by estimating percent change in CHD death rates using log-linear regression on time-series samples from the posterior distributions of the Bayesian model. To account for potential nonlinearity in trends, we calculated average annual percent change across four intervals: 1979–1990, 1990–2000, 2000–2010, and 2010–2017. These intervals roughly represent decades, with the last interval representing a period during which long term national declines in heart disease rates were interrupted and many counties experienced increasing heart disease death rates [3, 4, 10, 11].

**Suppression criteria.** For data for a given demographic group within a given county to be included, estimated rates were required to be reliable (i.e. the width of the credible interval must be less than the point estimate) and to have an annual population greater than 100 people for all years (1979–2017). This suppression criteria ensured that we only reported reliable rates in sufficiently large populations and that the same set of counties was used for the entire study period for each demographic group.

## Results

### National CHD death rates and trends

Between 1979 and 2017, national CHD death rates for all age, race, and gender groups declined substantially (Table 1). For all groups, national declines accelerated in the 2000s, but slowed during 2010–2017.

**Table 1. National age-standardized CHD death rates and annual percent change in CHD death rates by year, race, gender, and age group, United States, 1979–2017.**

| | Age group | | | Race (Ages 35 and older) | | | Gender (Ages 35 and older) | |
|---|---|---|---|---|---|---|---|---|
| | Ages 35 and older | Ages 35–64 | Ages 65 and older | White | Black | Other | Men | Women |
| CHD death rate (per 100,000) (95% CI) | | | | | | | | |
| 1979 | 662.4 (660.8, 664.1) | 160.1 (159.2, 161) | 2191.5 (2185.2, 2197.8) | 668.3 (666.5, 670.1) | 615.4 (609.8, 621.1) | 376.6 (363.1, 390.1) | 887.8 (884.7, 891) | 499.4 (497.5, 501.3) |
| 1990 | 487.5 (486.2, 488.8) | 104.3 (103.6, 105) | 1654.0 (1649.2, 1658.8) | 487.7 (486.3, 489.0) | 520.9 (516.2, 525.5) | 272.9 (265.2, 280.6) | 640.9 (638.4, 643.3) | 378.9 (377.5, 380.3) |
| 2000 | 364.7 (363.7, 365.7) | 74.8 (74.3, 75.4) | 1247.0 (1243.3, 1250.7) | 362.4 (361.4, 363.5) | 425.9 (422.2, 429.6) | 220.3 (215.4, 225.2) | 471.2 (469.4, 473.1) | 286.2 (285.1, 287.3) |
| 2010 | 221.6 (220.9, 222.3) | 52.2 (51.8, 52.6) | 737.3 (734.7, 739.9) | 221.5 (220.7, 222.2) | 255.5 (252.9, 258.1) | 139.3 (136.4, 142.2) | 294.9 (293.6, 296.1) | 165.7 (164.9, 166.5) |
| 2017 | 181.1 (180.5, 181.7) | 47.2 (46.9, 47.6) | 588.6 (586.4, 590.7) | 182.4 (181.8, 183.1) | 203.9 (201.8, 205.9) | 109.4 (107.4, 111.4) | 247.2 (246.1, 248.2) | 128.7 (128.1, 129.4) |
| Annual percent change (%) (95% CI) | | | | | | | | |
| 1979–1990 | -4.0 (-4.2, -3.7) | -3.0 (-3.2, -2.7) | -2.8 (-3.0, -2.5) | -3.0 (-3.3, -2.8) | -1.7 (-2.1, -1.4) | -2.0 (-2.8, -1.2) | -3.2 (-3.4, -2.9) | -2.7 (-3.0, -2.4) |
| 1990–2000 | -3.3 (-3.6, -3.0) | -2.8 (-3.0, -2.6) | -2.7 (-2.9, -2.5) | -2.8 (-3.1, -2.6) | -2.1 (-2.5, -1.8) | -2.2 (-2.7, -1.8) | -3.0 (-3.2, -2.8) | -2.7 (-2.9, -2.4) |
| 2000–2010 | -3.6 (-3.8, -3.4) | -5.1 (-5.4, -4.8) | -5.4 (-5.7, -5.1) | -5.0 (-5.3, -4.7) | -5.3 (-5.7, -4.9) | -4.7 (-5.0, -4.3) | -4.8 (-5.1, -4.5) | -5.6 (-5.9, -5.2) |
| 2010–2017 | -1.4 (-1.5, -1.3) | -2.8 (-3.2, -2.5) | -3.2 (-3.6, -2.8) | -2.7 (-3.1, -2.4) | -3.1 (-3.5, -2.7) | -3.3 (-4.2, -2.4) | -2.5 (-2.8, -2.1) | -3.5 (-3.9, -3.2) |

## County-level CHD death rates and trends by age group

The median county-level CHD death rates for each age group declined by roughly 3-fold from 1979 to 2017 (Table 2, Fig 1). In 1979, the spatial distributions of the highest CHD death rates differed across age groups, with the highest rates for ages 35–64 being concentrated in a band stretching from Kentucky and West Virginia through the Atlantic Coast and the highest rates for ages 65 and older being concentrated from Illinois through New York (Fig 2). By 2017, the geographic patterns for the two age groups were more similar, with the highest rates extending from West Virginia through New Mexico. One notable exception to these similar geographic patterns was a concentration of high CHD death rates in 2017 along the south Atlantic coast observed for ages 35–64 but not ages 65 and older.

Prior to 2000, all age groups experienced strong, widespread declines in CHD death rates, with few counties experiencing increasing death rates (Table 2, Fig 3, Fig 4). However, in 2000–2010, this pattern changed for ages 35–64, with some counties experiencing increases (11.3% of counties (95% CI: 9.7, 12.5)). In 2000–2010, most counties for ages 65 and older continued to decline, with only 1.1% of counties (95% CI: 0.8, 1.6)) experiencing increases. In 2010–2017, both age groups in many counties experienced increasing rates (40.5%, 95% CI: 38.4, 43.1 and 16.0%, 95% CI: 14.7, 17.2 for ages 35–64 and 65 and older, respectively). Prior to 2000, the geographic distributions of trends were similar for ages 35–64 and ages 65 and older (Fig 5). Slower declines and the few counties with increasing rates were concentrated in New Mexico, Oklahoma, and the Mississippi Delta for both ages 35–64 and 65 and older. However, after 2000, the geographic patterns for these age groups diverged, with increases becoming geographically widespread among ages 35–64 and remaining rare for ages 65 and older.

**Table 2. County-level distributions of age-standardized CHD death rates and annual percent change in CHD death rates by year, race, gender, and age group, United States, 1979–2017.**

| | Age group | | | Race (Ages 35 and older) | | | Gender (Ages 35 and older) | |
|---|---|---|---|---|---|---|---|---|
| | Ages 35 and older | Ages 35–64 | Ages 65 and older | White | Black | Other | Men | Women |
| Number of counties | 3114 | 3108 | 3097 | 3114 | 1594 | 857 | 3111 | 3106 |
| Median CHD death rate (per 100,000) (25th and 75th percentiles) | | | | | | | | |
| 1979 | 624.9 (555.3, 697.5) | 155.3 (134.0, 182.2) | 2033.5 (1812.0, 2275.0) | 631.3 (561.0, 707.0) | 605.5 (526.2, 697.4) | 242.7 (186.4, 328.6) | 850.7 (758.1, 944.3) | 450.5 (393.4, 514.5) |
| 1990 | 466.1 (411.2, 524.9) | 105.9 (88.5, 126.9) | 1556.1 (1369.0, 1746.7) | 467.3 (411.5, 525.2) | 509.6 (440.2, 584.3) | 241.9 (195.2, 312.3) | 625.6 (553.1, 702.3) | 348.5 (301.3, 398.8) |
| 2000 | 352.2 (297.8, 409.9) | 78.8 (63.8, 98.2) | 1177.5 (1001.7, 1368.0) | 352.8 (298.2, 408.0) | 404.3 (345.2, 477.6) | 196.3 (157.5, 259.7) | 466.9 (398.9, 532.0) | 267.6 (220.1, 318.4) |
| 2010 | 224.1 (187.4, 269.5) | 58.3 (46.4, 76.3) | 719.7 (605.8, 863.8) | 222.8 (186.6, 269.0) | 242.0 (202.1, 294.6) | 131.3 (104.4, 175.4) | 302.9 (256.8, 356.1) | 163.4 (133.5, 202.4) |
| 2017 | 193.1 (159.5, 233.7) | 55.8 (42.1, 74.0) | 604.8 (507.9, 724.1) | 192.3 (159.7, 233.5) | 205.3 (167.4, 254.8) | 103.3 (83.4, 145.7) | 266.0 (222.3, 314.9) | 132.1 (106.7, 165.1) |
| Median annual percent change (%) (25th and 75th percentiles) | | | | | | | | |
| 1979–1990 | -2.8 (-3.6, -1.9) | -3.7 (-4.5, -2.7) | -2.6 (-3.5, -1.7) | -2.9 (-3.7, -2.0) | -1.7 (-3, -0.3) | 0.1 (-1.4, 1.6) | -2.9 (-3.7, -2.1) | -2.5 (-3.4, -1.5) |
| 1990–2000 | -2.8 (-3.7, -1.9) | -3.0 (-3.9, -2.0) | -2.7 (-3.7, -1.8) | -2.8 (-3.7, -1.9) | -2.3 (-3.4, -1.2) | -2.1 (-3.6, -0.8) | -2.9 (-3.7, -2.1) | -2.6 (-3.7, -1.5) |
| 2000–2010 | -4.6 (-5.6, -3.4) | -3.0 (-4.2, -1.6) | -4.9 (-6.0, -3.8) | -4.5 (-5.6, -3.4) | -5.1 (-6.4, -3.8) | -4.2 (-5.6, -3.0) | -4.4 (-5.3, -3.3) | -4.9 (-6.1, -3.6) |
| 2010–2017 | -2.1 (-3.4, -0.8) | -0.7 (-2.1, 0.8) | -2.5 (-3.8, -1.1) | -2.2 (-3.4, -0.8) | -2.3 (-3.8, -0.8) | -2.8 (-4.1, -1.5) | -1.9 (-3, -0.6) | -2.9 (-4.4, -1.2) |

## County-level CHD death rates and trends by race

Across the study period, the distributions of county-level rates for blacks and whites, ages 35 and older, were similar (Table 2, Fig 1). On average, blacks had the highest CHD death rates, followed by whites, with other races having the lowest rates. The highest rates for whites and blacks exhibited similar spatiotemporal trends, with the highest CHD death rates starting in the Northeast in 1979 and moving to a band stretching across Texas, Arkansas, and Louisiana by 2017 (Fig 6).

Compared to whites, the distributions of trends for blacks and other races have shifted from having slower declines and some counties with increasing rates during 1979–1990 to having similar distributions of percent change since 2010 (Table 2, Fig 3). Likewise, the percentages of counties experiencing increases was higher among blacks and other races compared to whites prior to 2010, but there was no difference after 2010 (Table 2, Fig 4) (20.0% (95% CI: 18.7, 21.8), 20.6% (95% CI: 18.8, 23.7), 17.4% (95% CI: 13.2, 25.1) for whites, blacks, and other races, respectively). Prior to 2000, for blacks and whites, the slowest declines and increases were concentrated in counties stretching across East Texas, Oklahoma, and the Mississippi Delta (Fig 7). However, during 2010–2017, the geographic patterns of slow declines and increases were widespread for all races.

## County-level CHD Death Rates and Trends by Gender

By gender, the distributions of county-level death rates were almost completely distinct in 1979, but the overlap of the distributions increased over time (Table 2, Fig 1). The geographic patterns of high CHD death rates were similar for both genders, with the highest rates initially stretching from Illinois through New York and ending the study period in a band from West Virginia through New Mexico (Fig 8).

Distributions of percent change were similar across genders prior to 2000, but shifted after 2000 such that, on average, men declined more quickly than women (Table 2, Fig 3). Likewise,

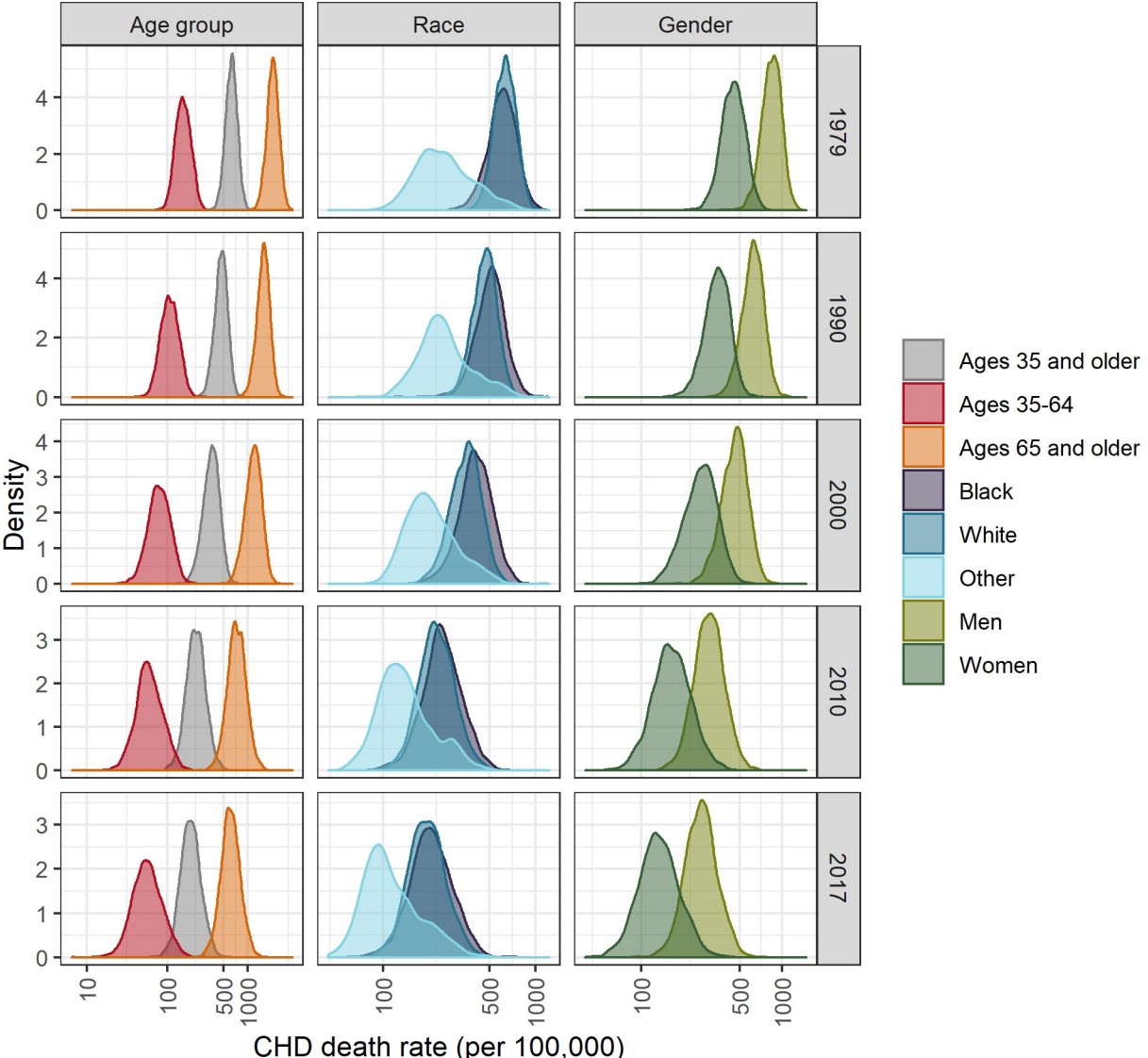

**Fig 1. Distributions of county-level CHD death rates by year, age group, race, and gender, United States, 1979–2017.** Race and gender groups are ages 35 and older.

prior to 2010 increasing county-level CHD death rates were more common among women but were recently more common among men (Fig 4). The geographic patterns of trends for both genders were similar for all time periods (Fig 9).

## Discussion

Using a spatiotemporal Bayesian model, we estimated county-level CHD death rates and trends over an uninterrupted 39-year period. These methods permitted the examination of trends in counties across the United States by age, race, and gender, including counties with small populations and small numbers of deaths. These results provide detailed documentation of county-level patterns of CHD death rates and trends that have been masked by previous national analyses [1, 5–8].

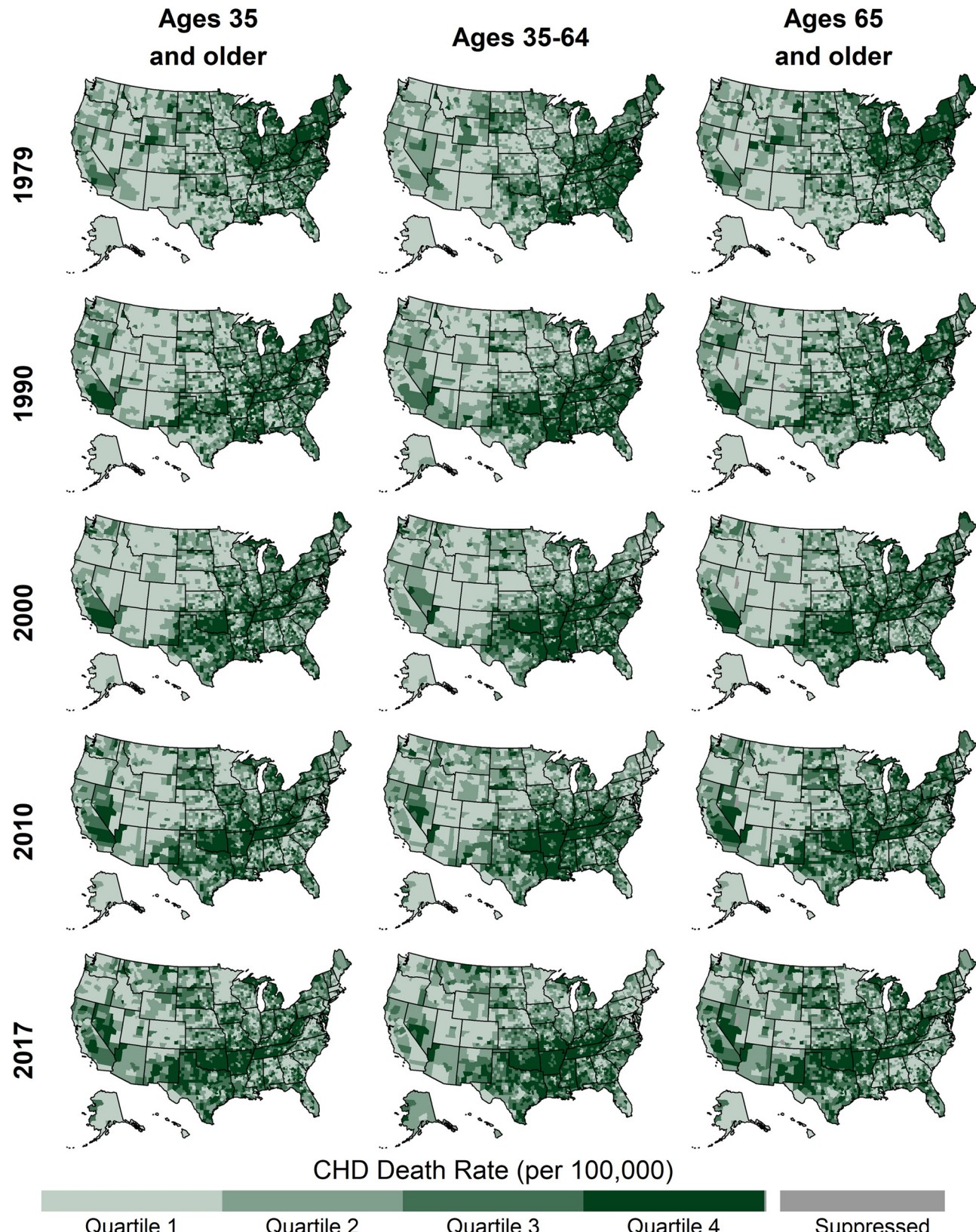

**Fig 2. County-level CHD death rates by year and age group, United States, 1979–2017.** Quartile ranges for each map are shown in Table 2.

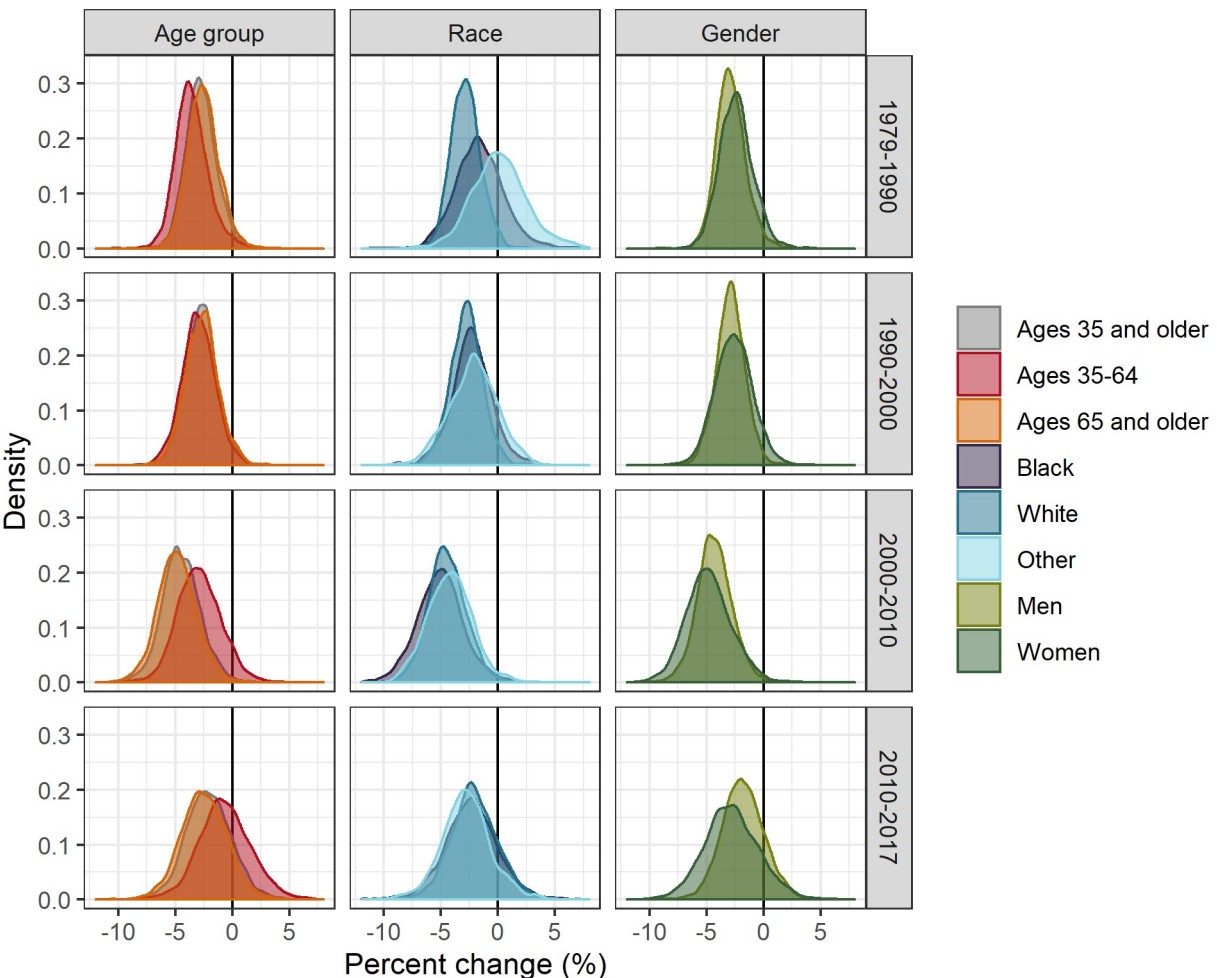

**Fig 3. Distributions of percent change in county-level CHD death rates by time period, age group, race, and gender, United States, 1979–2017.** Race and gender groups are ages 35 and older.

Across age group, race, and gender, the highest CHD disease death rates became concentrated in a band stretching from West Virginia through east Texas. The similarities in these patterns across demographic groups highlight the importance of place with respect to the burden of CHD mortality [6, 24]. By documenting these patterns through maps, this paper provides critical information for understanding the geographic patterns of CHD death rates for each demographic group.

Between 1979 and 2017, county-level CHD death rates declined substantially across age, race, and gender. However, even during this period of broad national declines, non-white populations, working age adults, and counties in the Mississippi Delta, Oklahoma, East Texas, and New Mexico experienced increases. Recently, these patterns have broadened such that declines have slowed across all age, race, and gender groups and have begun to increase in counties across the country.

The evolution of these trends over time by age, race, and gender suggests potential drivers of these changes. The pattern of trends by age group, with stagnating and increasing trends initially concentrated among ages 35–64, but recently common across age groups, suggests a birth cohort effect in which younger people have lived with CHD risk factors for more of their lives [11, 25, 26]. A similar pattern has been previously observed for deaths due to all diseases

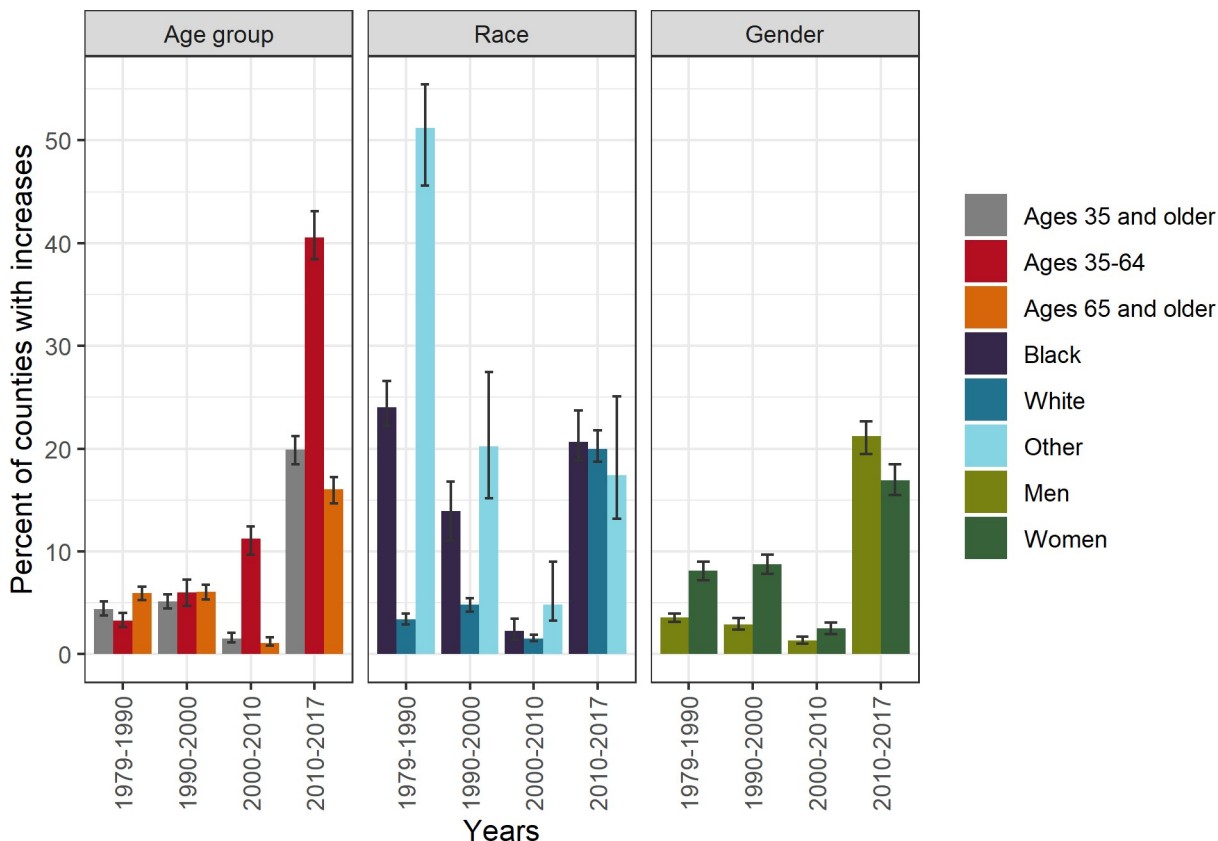

**Fig 4. Percent of counties with increasing CHD death rates by time period, age group, race, and gender, United States, 1979–2017.** Race and gender groups are ages 35 and older.

of the heart [11]. Additionally, the early pattern of trends by race, with county-level increases and slow declines initially concentrated in non-white populations prior to 2000, supports prior work on the timing of declines in CHD mortality among non-white populations. Although national declines in CHD mortality began in the 1960s, strong declines in CHD mortality among non-white populations were delayed by roughly a decade[25, 27–29]. These delayed declines could account for the large proportion of counties with increasing death rates for other races before 1990 [29]. Finally, the differences by gender, with initial increases in CHD death rates being rare but more prevalent among women while recent increases are more common and more prevalent among men, may reflect established gender differences in CHD risk factors, prevention, treatment, and health behaviors and changes in these factors over time [30–33].

Additionally, the temporal evolution of county-level geographic patterns demonstrates the influence of both traditional risk factors and social determinants of health. Across all demographic groups, increases and slow declines prior to 2010 were concentrated across the South in the Mississippi Delta, east Texas and Oklahoma. Although this geographic pattern is similar to that of traditional risk factors like obesity and diabetes, it also aligns with the geographic patterns of many social determinants of health like poverty, health insurance coverage, and educational attainment [34–36].

Within the context of recent increases in other cardiovascular disease outcomes, including deaths due to stroke, all diseases of the heart, and heart failure [1, 3, 4, 8, 17, 37–39],

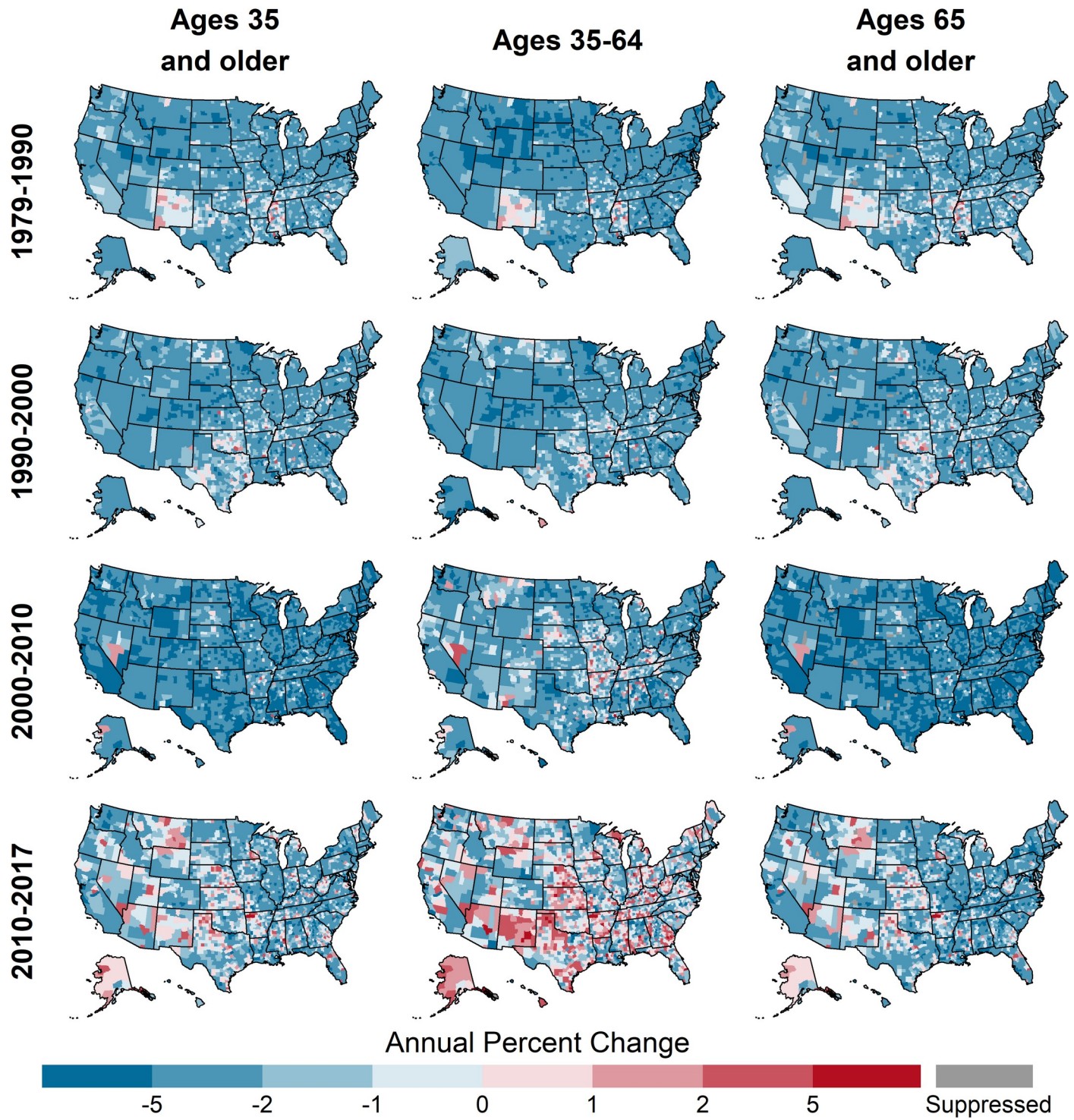

**Fig 5. Annual percent change in county-level CHD death rates by year and age group, United States, 1979–2017.**

documenting these recent county-level increases in CHD death rates provides key information that can inform responses by communities, public health professionals, clinicians, and researchers. For communities, these county-level data are critical to enhancing their

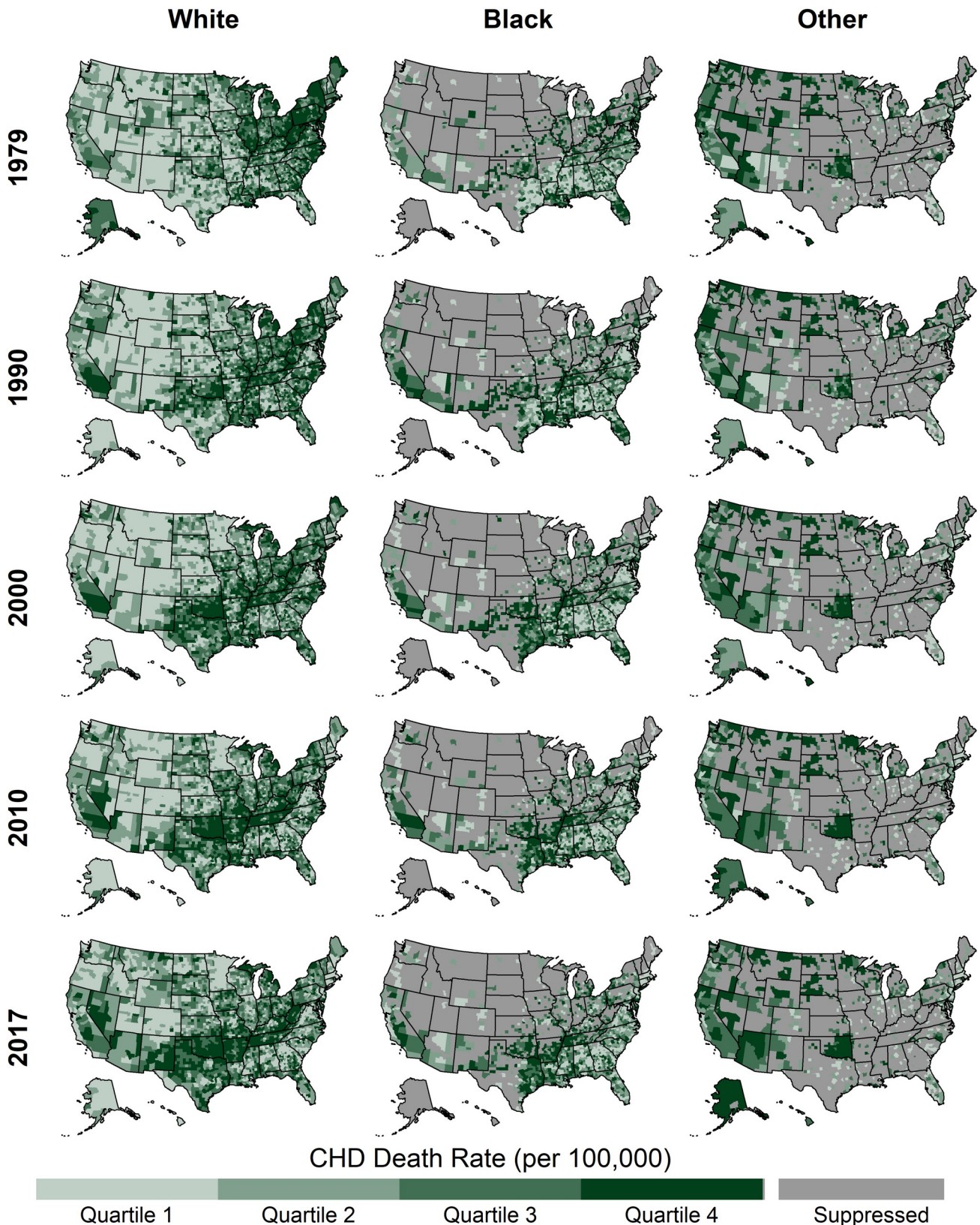

CHD Death Rate (per 100,000)

Quartile 1　　Quartile 2　　Quartile 3　　Quartile 4　　Suppressed

**Fig 6. County-level CHD death rates by year and race, ages 35 and older, United States, 1979–2017.** Quartile ranges are shown in Table 2.

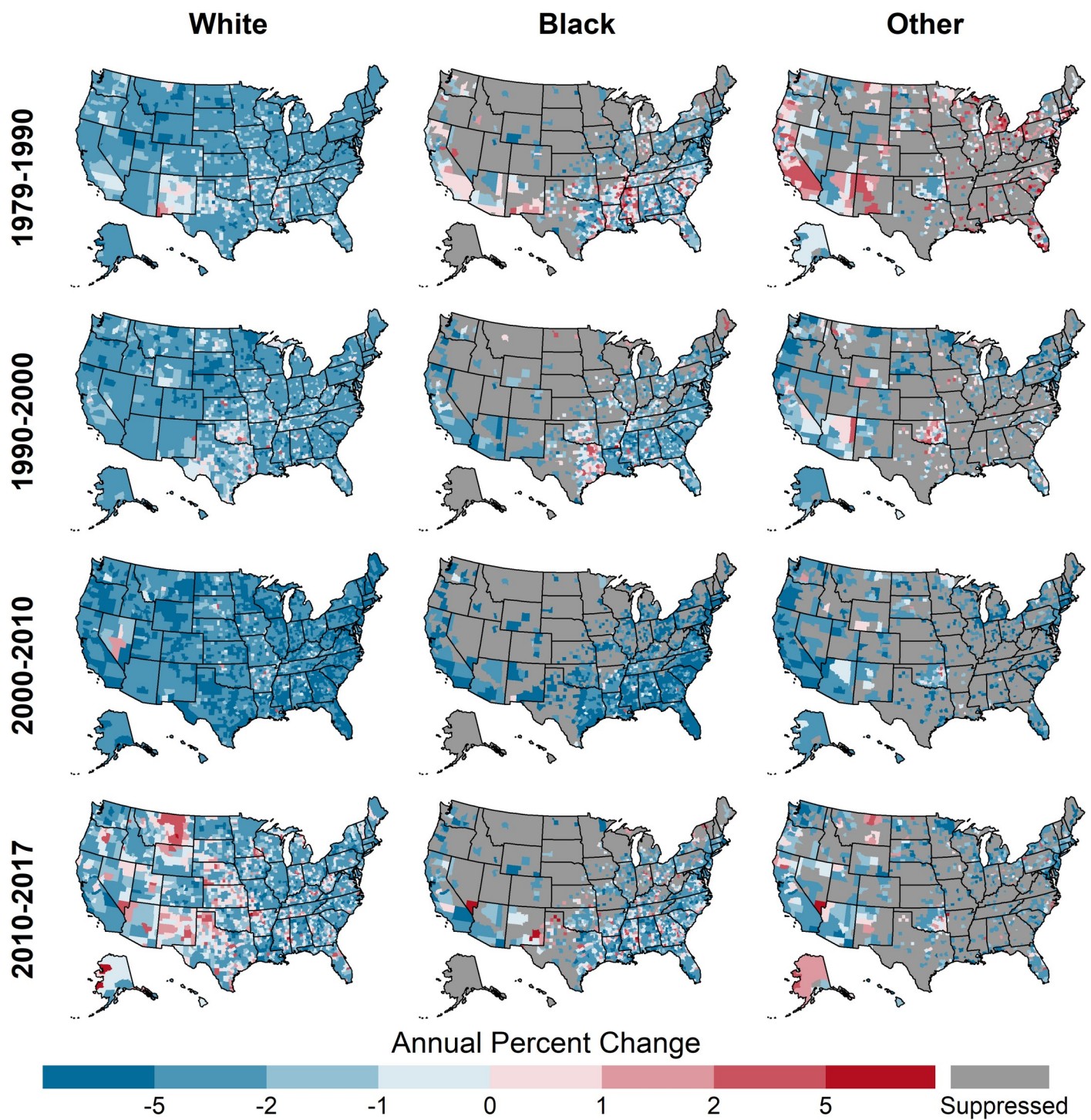

**Fig 7. Annual percent change in county-level CHD death rates by year and race, ages 35 and older, United States, 1979–2017.**

understanding of their health landscape and provide data to improve advocacy for health promotion.

For the public health and medical communities, our results identify where responses to the recent troubling trends are most needed [40]. These responses could address CHD prevention

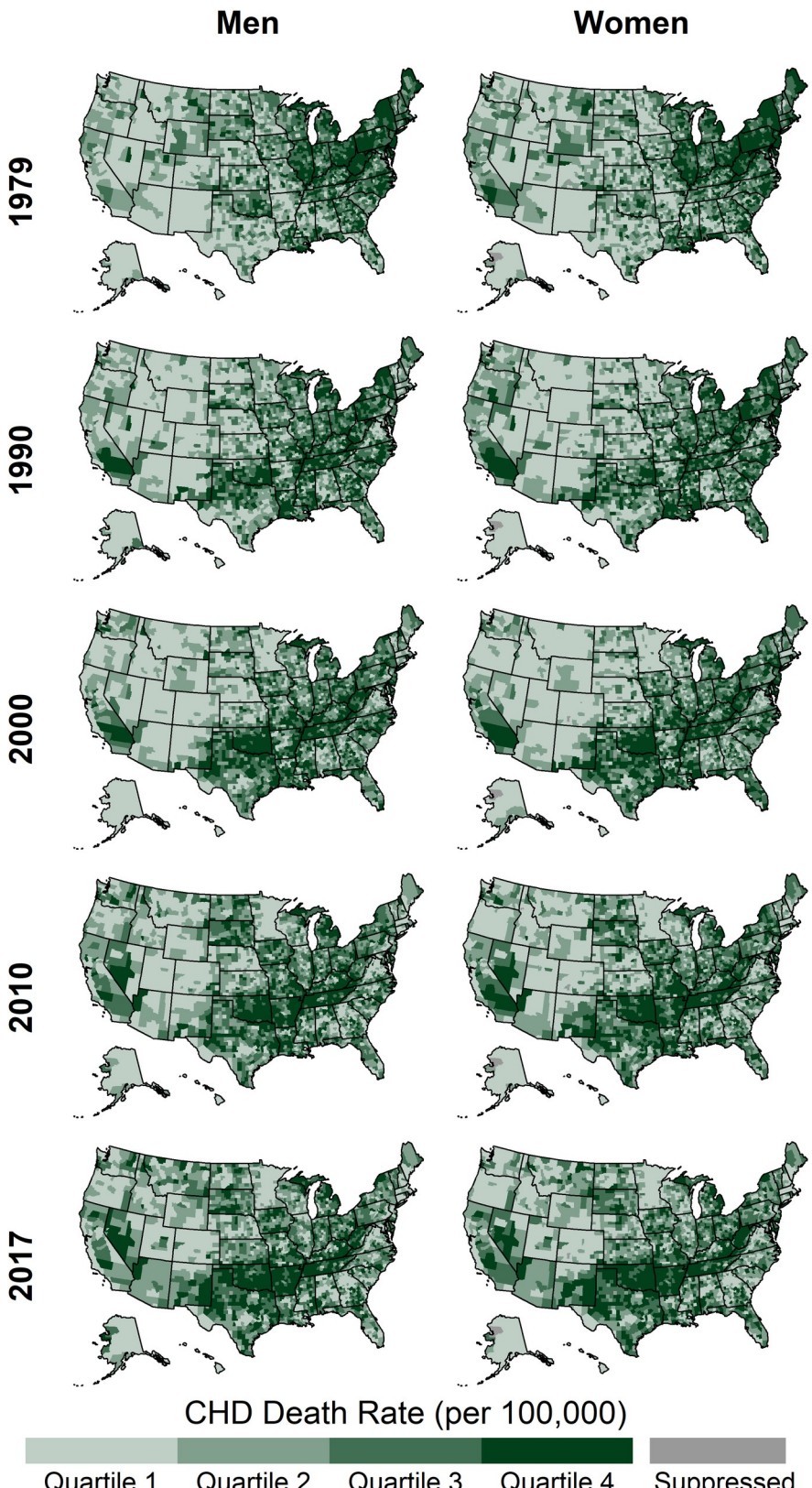

**Fig 8. County-level CHD death rates by year and gender, ages 35 and older, United States, 1979–2017.** Quartile ranges are shown in Table 2.

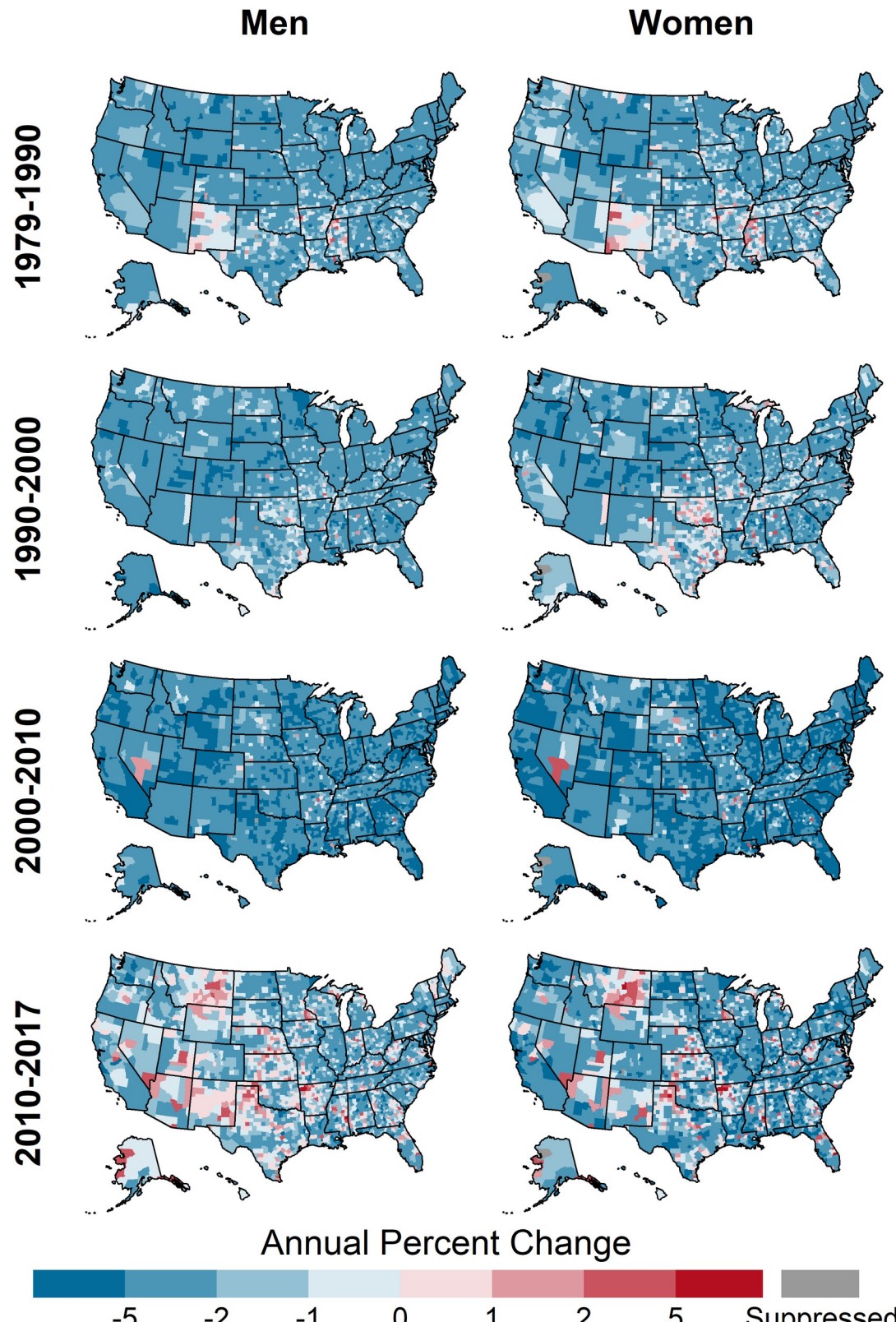

**Fig 9. Annual percent change in county-level CHD death rates by year and gender, ages 35 and older, United States, 1979–2017.**

and treatment among younger adults, especially given this group's increasing prevalence of cardiovascular disease risk factors [41]. Also, the public health and medical communities could benefit from defining and implementing structural interventions to address place-based characteristics (e.g., social determinants of health, the built environment, policies) [42–44], which are critical given the role of these characteristics in establishing individual-level cardiovascular disease risk factors [24, 35, 36, 45]. Finally, these mortality data also point to the need for greater surveillance of CHD, including its incidence, risk factors, and data to support geographic analyses [46].

In addition to robust public health and clinical responses to these recent increases, additional research is needed to better understand these local trends in CHD death rates. First, our county-level estimates (see supplement for data) could be used to investigate local drivers of changing CHD death rates. Nationally, primary prevention accounts for 50% of the declines in CHD mortality from 1980 through 2000, with secondary prevention accounting for the other 50% [47]. However, these relative contributions are likely to vary geographically and over time, especially in light of the geographic variations in the slowing, plateauing, and increasing trends in CHD death rates observed in this study. Additionally, the locations and populations in which early slower declines and increases were concentrated suggest the importance of looking upstream to social factors, economic factors, and policies that can subsequently drive changes in proximal biological and behavioral risk factors and, eventually, changes in CHD death rates [48–50].

A key strength of this study is its application of a fully Bayesian spatiotemporal model to national surveillance data. By borrowing statistical strength across adjacent counties, demographic groups, and years, this model estimated precise, reliable rates, even in counties with small numbers of deaths, thereby permitting the inclusion of more counties than would be possible with other methods [21]. Additionally, this analysis used national vital statistics data that included all recorded deaths over an uninterrupted time period, allowing the comparison of rates and trends over demographic groups and time. Finally, by using linear regression within the Bayesian results, our percent change estimates account for imprecision in underlying CHD death rates.

Temporal changes in the definition of CHD on the death certificate and initial absence of ethnicity classifications represent key limitations of this long-term spatiotemporal analysis. Our study was limited to beginning in 1979 because of poor comparability ratios for CHD deaths prior to ICD-9 [51]. However, even within this period with consistency in comparability rations, misclassification of CHD on death certificates presents another possible limitation. Although there have been few studies of temporal trends in differential misclassification by demographic groups, especially with respect to recent changes in CHD mortality trends, past work has found lower accuracy of CHD as a cause of death in older adults [52]. However, this misclassification is unlikely to fully explain the observed trends. Also, although the length of the study allowed the examination of long-term trends, it limited the ability to examine trends for Hispanic ethnicity or for races other than black and white because Hispanic ethnicity was routinely collected on death certificates only after 1996 [14].

## Conclusion

In this analysis of county-level CHD death rates from 1979 through 2017, we documented spatiotemporal differences over age, race, and gender in both the level of the CHD death rates and their trends. Specifically, county-level increases in CHD death rates were uncommon prior to 2010, but were common and widespread after 2010 across age, race, and gender, especially among working age adults. These county-level results can inform critical responses by the

public health community, medical providers, researchers, and communities to address troubling recent trends.

## Supporting information

**S1 File. Compressed file of county-level CHD death rates and trends.**
(ZIP)

## Acknowledgments

The findings and conclusions in this report are those of the authors and do not necessarily represent the official position of the Centers for Disease Control and Prevention.

## Author Contributions

**Conceptualization:** Adam S. Vaughan, Linda Schieb, Michele Casper.

**Data curation:** Adam S. Vaughan.

**Formal analysis:** Adam S. Vaughan.

**Investigation:** Adam S. Vaughan.

**Methodology:** Adam S. Vaughan, Linda Schieb.

**Project administration:** Michele Casper.

**Resources:** Adam S. Vaughan.

**Supervision:** Michele Casper.

**Validation:** Adam S. Vaughan.

**Visualization:** Adam S. Vaughan.

**Writing – original draft:** Adam S. Vaughan.

**Writing – review & editing:** Adam S. Vaughan, Linda Schieb, Michele Casper.

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
