## [Decision Letter · Decision Letter 0]

15 Apr 2020

PONE-D-20-04975

Historic and recent trends in county-level coronary heart disease death rates by race, gender, and age group, United States, 1979-2017

PLOS ONE

Dear Dr  Adam Vaughan,

Thank you for submitting your manuscript to PLOS ONE. After careful consideration, we feel that it has merit but does not fully meet PLOS ONE’s publication criteria as it currently stands. Therefore, we invite you to submit a revised version of the manuscript that addresses the points raised during the review process.

We were very happy to read your article.  There are, however, some issues the reviewers need you to clarify/add before the manuscript can be accepted to be published.

We would appreciate receiving your revised manuscript by May 30 2020 11:59PM. To enhance the reproducibility of your results, we recommend that if applicable you deposit your laboratory protocols in protocols.io, where a protocol can be assigned its own identifier (DOI) such that it can be cited independently in the future. For instructions see: http://journals.plos.org/plosone/s/submission-guidelines#loc-laboratory-protocols

We look forward to receiving your revised manuscript.

Kind regards,

Katriina Aalto-Setala, Professor

Academic Editor

PLOS ONE

Journal Requirements:

2. We note that Figures 2, 5, 6, 7, 8 and 9 in your submission contain [map/satellite] images which may be copyrighted. All PLOS content is published under the Creative Commons Attribution License (CC BY 4.0), which means that the manuscript, images, and Supporting Information files will be freely available online, and any third party is permitted to access, download, copy, distribute, and use these materials in any way, even commercially, with proper attribution. For these reasons, we cannot publish previously copyrighted maps or satellite images created using proprietary data, such as Google software (Google Maps, Street View, and Earth). For more information, see our copyright guidelines: http://journals.plos.org/plosone/s/licenses-and-copyright.

a)    You may seek permission from the original copyright holder of Figures 2, 5, 6, 7, 8 and 9 to publish the content specifically under the CC BY 4.0 license.  

Reviewers' comments:

Reviewer's Responses to Questions

**Comments to the Author**

1. Is the manuscript technically sound, and do the data support the conclusions?

Reviewer #1: Yes

Reviewer #2: Partly

Reviewer #3: Yes

2. Has the statistical analysis been performed appropriately and rigorously? 

Reviewer #1: Yes

Reviewer #2: I Don't Know

Reviewer #3: Yes

3. Have the authors made all data underlying the findings in their manuscript fully available?

Reviewer #1: Yes

Reviewer #2: Yes

Reviewer #3: Yes

4. Is the manuscript presented in an intelligible fashion and written in standard English?

Reviewer #1: Yes

Reviewer #2: Yes

Reviewer #3: Yes

5. Review Comments to the Author

Reviewer #1: The following reflects a biostatistical review of the manuscript.

In this work, the authors perform a thorough examination of geographic variation in coronary heart disease (CHD) mortality, and in temporal trends for CHD mortality. They used a Bayesian multivariate space-time autoregressive model, allowing for "borrowing strength" in estimation, for neighboring counties, years, and demographic groups. Specifically, a strength of this method is that it allows for stable estimation of rates and trends even in years/ages/counties/demographic groups with very few observations. The authors identified significant spatiotemporal variation in CHD rates. In particular, they identified counties with recent increases in CHD mortality, going against the overall national trend of CHD mortality reduction.

While this manuscript is excellent overall, I believe it would benefit from some relatively minor changes. In particular, since the modeling is such a critical part of the work, it would greatly strengthen the manuscript to include a more detailed description of the model details. The results are difficult to interpret without the context of the model details, and it is appropriate to include those details within this manuscript rather than requiring the reader to look elsewhere.

The methods section (line 71) states that the model is able to borrow strength across adjacent demographic groups. How is adjacency defined in this context? Aside from adjacency between age groups, adjacency relationships (and the strength of these relationships) between demographic groups are not obvious. Please include details on this in the methods section.

How were the four intervals (1979-1990, 1990-2000, 2000-2010, 2010-2017) chosen for the APC calculations?

Reviewer #2: Vaughan et al examined the trends of county-level death rates due to coronary heart disease in the US from 1797 to 2017. My comments are listed below:

1. Line 65: in this section it is not clear whether the death rates were crude rates or standardized/adjusted rates. From the tables presented it seems that the rates were age standardized. If this is the case, please specify it in the methods.

2. Line 77: Please provide rationale for only including death at age 35 and above? And the reason of categorizing age groups as 35-64, ages 65 and older?

3. Line 233-238: The authors should consider the possibility of changes in the validity of death certificate coding during the study period. With greater focus on population levels of blood pressure its plausible that the validity of codes assigning cause of death to coronary heart disease may have not been stable over this time period. And how this can affect the results.

Reviewer #3: This is a well-written, important manuscript that provides key data on changes in CHD mortality rates in the US over time. There are additional analyses/considerations that could strengthen the findings of this paper.

1. In the introduction, there is no acknowledgement of work on premature CVD mortality as recently published by Chen and colleagues (JAMA Cardiology doi:10.1001/jamacardio.2019.3891). This work should be cited and briefly described.

2. Why were those aged 35-64 combined into one group? CHD events before age 50 may be different from those after age 50 (could be more related to genetic susceptibility). Were there considerations about dividing up the groups 35-49 vs. 50-64 vs. 65+?

6. PLOS authors have the option to publish the peer review history of their article (what does this mean?). If published, this will include your full peer review and any attached files.

Reviewer #1: No

Reviewer #2: No

Reviewer #3: No

---

## [Author Response · Author response to Decision Letter 0]

11 May 2020

Thank you for the thorough review of this paper. Our responses to each comment are presented below.

Reviewer #1: The following reflects a biostatistical review of the manuscript.

In this work, the authors perform a thorough examination of geographic variation in coronary heart disease (CHD) mortality, and in temporal trends for CHD mortality. They used a Bayesian multivariate space-time autoregressive model, allowing for "borrowing strength" in estimation, for neighboring counties, years, and demographic groups. Specifically, a strength of this method is that it allows for stable estimation of rates and trends even in years/ages/counties/demographic groups with very few observations. The authors identified significant spatiotemporal variation in CHD rates. In particular, they identified counties with recent increases in CHD mortality, going against the overall national trend of CHD mortality reduction.

While this manuscript is excellent overall, I believe it would benefit from some relatively minor changes. In particular, since the modeling is such a critical part of the work, it would greatly strengthen the manuscript to include a more detailed description of the model details. The results are difficult to interpret without the context of the model details, and it is appropriate to include those details within this manuscript rather than requiring the reader to look elsewhere.

Response: We have modified the methods (page 5) to include much greater detail about the model.

The methods section (line 71) states that the model is able to borrow strength across adjacent demographic groups. How is adjacency defined in this context? Aside from adjacency between age groups, adjacency relationships (and the strength of these relationships) between demographic groups are not obvious. Please include details on this in the methods section.

Response: We have included these details in the methods (page 5).

How were the four intervals (1979-1990, 1990-2000, 2000-2010, 2010-2017) chosen for the APC calculations?

Response: We have included a description of how those intervals were chosen in the methods (line 108).

Reviewer #2: Vaughan et al examined the trends of county-level death rates due to coronary heart disease in the US from 1797 to 2017. My comments are listed below:

1. Line 65: in this section it is not clear whether the death rates were crude rates or standardized/adjusted rates. From the tables presented it seems that the rates were age standardized. If this is the case, please specify it in the methods.

Response: All death rates were age-standardized using the 2000 US standard population. This was previously included in the manuscript (line 83), so no change has been made.

2. Line 77: Please provide rationale for only including death at age 35 and above? And the reason of categorizing age groups as 35-64, ages 65 and older?

Response: Deaths were only included for ages 35 and older because deaths from CHD below age 35 are exceedingly rare. Including these deaths would therefore add to the denominator but not the numerator, lowering death rates in this middle-aged group. We selected these two age groups based on both past work and on public health programs. Our prior work found that ages 35-64 have experienced widespread increases in death rates due to cardiovascular disease, but these death rates have continued to decline for ages 65 and older. Additionally, national initiatives such as Million Hearts include adults ages 35-64 as a target population. Our inclusion of this age group in this paper therefore directly informs public health practice. We have clarified this in the introduction (line 47) and in the methods (line 70).

3. Line 233-238: The authors should consider the possibility of changes in the validity of death certificate coding during the study period. With greater focus on population levels of blood pressure its plausible that the validity of codes assigning cause of death to coronary heart disease may have not been stable over this time period. And how this can affect the results.

Response: We have included this as a limitation (line 263).

Reviewer #3: This is a well-written, important manuscript that provides key data on changes in CHD mortality rates in the US over time. There are additional analyses/considerations that could strengthen the findings of this paper.

1. In the introduction, there is no acknowledgement of work on premature CVD mortality as recently published by Chen and colleagues (JAMA Cardiology doi:10.1001/jamacardio.2019.3891). This work should be cited and briefly described.

Response: Thank you for pointing out this key reference. We have included this reference and its context in the introduction (line 42). 

2. Why were those aged 35-64 combined into one group? CHD events before age 50 may be different from those after age 50 (could be more related to genetic susceptibility). Were there considerations about dividing up the groups 35-49 vs. 50-64 vs. 65+?

Response: Deaths were only included for ages 35 and older because deaths from CHD below age 35 are exceedingly rare. Including these deaths would therefore add to the denominator but not the numerator, lowering death rates in this middle-aged group. We selected these two age groups based on both past work and on public health programs. Our prior work found that ages 35-64 have experienced widespread increases in death rates due to cardiovascular disease, but these death rates have continued to decline for ages 65 and older. Additionally, national initiatives such as Million Hearts include adults ages 35-64 as a target population. Our inclusion of this age group in this paper therefore directly informs public health practice. We have clarified this in the introduction (line 47) and in the methods (line 70).

---

## [Decision Letter · Decision Letter 1]

24 Jun 2020

Historic and recent trends in county-level coronary heart disease death rates by race, gender, and age group, United States, 1979-2017

PONE-D-20-04975R1

Dear Dr. Vaughan,

We’re pleased to inform you that your manuscript has been judged scientifically suitable for publication and will be formally accepted for publication once it meets all outstanding technical requirements.

Kind regards,

Katriina Aalto-Setala, Professor

Academic Editor

PLOS ONE

Additional Editor Comments (optional):

Reviewers' comments:

Reviewer's Responses to Questions

**Comments to the Author**

1. If the authors have adequately addressed your comments raised in a previous round of review and you feel that this manuscript is now acceptable for publication, you may indicate that here to bypass the “Comments to the Author” section, enter your conflict of interest statement in the “Confidential to Editor” section, and submit your "Accept" recommendation.

Reviewer #1: All comments have been addressed

Reviewer #2: All comments have been addressed

Reviewer #3: All comments have been addressed

2. Is the manuscript technically sound, and do the data support the conclusions?

Reviewer #1: Yes

Reviewer #2: Yes

Reviewer #3: Yes

3. Has the statistical analysis been performed appropriately and rigorously? 

Reviewer #1: Yes

Reviewer #2: Yes

Reviewer #3: Yes

4. Have the authors made all data underlying the findings in their manuscript fully available?

Reviewer #1: Yes

Reviewer #2: Yes

Reviewer #3: Yes

5. Is the manuscript presented in an intelligible fashion and written in standard English?

Reviewer #1: Yes

Reviewer #2: Yes

Reviewer #3: Yes

6. Review Comments to the Author

Reviewer #1: (No Response)

Reviewer #2: (No Response)

Reviewer #3: All comments have been addressed adequately. There are no further comments at this time. The authors have also addressed the questions regarding the statistical methods.

7. PLOS authors have the option to publish the peer review history of their article (what does this mean?). If published, this will include your full peer review and any attached files.

Reviewer #1: No

Reviewer #2: No

Reviewer #3: No

---

## [Editor Report · Acceptance letter]

25 Jun 2020

PONE-D-20-04975R1 

Historic and recent trends in county-level coronary heart disease death rates by race, gender, and age group, United States, 1979-2017 

Dear Dr. Vaughan:

I'm pleased to inform you that your manuscript has been deemed suitable for publication in PLOS ONE. Congratulations! Your manuscript is now with our production department. 

Kind regards, 

on behalf of

Dr Katriina Aalto-Setala 

Academic Editor

PLOS ONE